# Association between 10-Year Atherosclerotic Cardiovascular Disease Risk and Estimated Glomerular Filtration Rate in Chinese People with Normal to Slightly Reduced Kidney Function: A Cross-Sectional Study

**DOI:** 10.3390/ijerph192316300

**Published:** 2022-12-05

**Authors:** Feilong Chen, Junting Liu, Shaomei Han, Tao Xu

**Affiliations:** 1Department of Epidemiology and Statistics, Institute of Basic Medical Sciences, Chinese Academy of Medical Sciences & School of Basic Medicine, Peking Union Medical College, Beijing 100005, China; 2Child Health Big Data Research Center, Capital Institute of Pediatrics, Beijing 100020, China

**Keywords:** estimated glomerular filtration rate, atherosclerotic cardiovascular diseases, Chinese, kidney function

## Abstract

Many studies suggest that cardiovascular-related mortality is higher in patients with end-stage renal disease, but few focus on the risk of atherosclerotic cardiovascular disease (ASCVD) in subjects with normal to slightly reduced kidney function. Our study aimed to explore the association between normal to slightly decreased estimated glomerular filtration rate (eGFR) and 10-year ASCVD risk levels among subjects with relative health conditions. A total of 12,986 subjects from the Chinese Physiological Constant and Health Condition survey were included. We used the Chronic Kidney Disease Epidemiology Collaboration equations to calculate eGFR and the 10-year ASCVD risk scores to assess the subjects’ risk of 10-year ASCVD. Ordinal logistic regression analysis was conducted to explore the association between ASCVD risk levels and eGFR, and adjust the possible confounding factors. Results indicated that the 10-year ASCVD risk scores gradually increased following the decrease in eGFR. Subjects who had smaller eGFR were more likely to have a greater risk of 10-year ASCVD. Additionally, the association between eGFR and 10-year ASCVD risk level changed with varying eGFR. The risk of one or more levels increasing in the 10-year ASCVD risk group was 5.20 times (Quartile 2 [Q2], 95%CI: 3.90, 6.94), 9.47 times (Q3, 95%CI: 7.15, 12.53) and 11.41 times (Q4, 95%CI: 8.61, 15.12) higher compared with Q1. We found that eGFR was significantly associated with 10-year ASCVD risk among Chinese subjects with normal to slightly reduced kidney function. Therefore, clinicians should strengthen the monitoring of cardiovascular risk in patients with renal impairment (even if renal function is only mildly reduced), assess the patients’ risk of ASCVD, and take early action in high-risk groups to reduce the risk of adverse atherosclerotic cardiovascular events.

## 1. Introduction

Chronic kidney disease (CKD) was defined as chronic structural and functional abnormalities of the kidneys, affecting approximately 8–16% of the world’s population [1]. Globally, CKD’s prevalence was 9.1% in 2017, with a 41.5% increase in deaths since 1990 [2]. A meta-analysis, which focused on the CKD prevalence in Chinese adults and the elderly, showed that the unstandardized rates of CKD were 13.39% in Chinese adults and 19.25% in the elderly aged over 60, which were much higher than in developed countries and the global average level [1]. On the other hand, studies have also revealed that high CKD prevalence is associated with an increased incidence of end-stage renal disease (ESRD) and mortality from cardiovascular disease (CVD) [3], putting a heavy burden on economic and social development. Several studies have indicated that patients with ESRD have a CVD mortality that is 10 to 20 times greater than the general population [4,5,6]. Moreover, the National Kidney Foundation (NKF) Task Force on Cardiovascular Disease in Chronic Renal Disease issued a report in 1998, emphasizing that the high risk of CVD in patients with CKD [7]. This report showed that there was a high prevalence of CVD in CKD and that mortality due to CVD was 10 to 30 times higher in dialysis patients than in the general population [8,9]. The task force therefore recommended that patients with CKD should be considered in the “highest risk group” for subsequent CVD events and that treatment based on CVD risk stratification should take into account the highest risk status of patients with CKD [5].

Glomerular filtration rate (GFR) is a common clinical measure of renal function. The staging of GFR can be seen in the guideline [10]. Direct measurement of GFR could be achieved by calculating the clearance of agents like iohexol or iothalamate [11,12,13]; however, the update of estimating equations (e.g., the Chronic Kidney Disease Epidemiology Collaboration [CKD-EPI] [14,15] and Modification of Diet in Renal Disease Study [MDRD] [16] equations) has largely simplified the measurement in clinical practice. The United States and much of the world prefer the CKD-EPI equation, which is more accurate than the MDRD ones, particularly for subjects with eGFR values greater than 60 mL/min/1.73 m^2^ [14,16].

Atherosclerotic cardiovascular disease (ASCVD), defined as nonfatal acute myocardial infarction (MI), coronary heart disease (CHD) death, or fatal or nonfatal stroke [17], is a common type of CVD. There are several cardiovascular disease prediction models available worldwide, including the Framingham general CVD equations in the United States [18] and the Systematic Coronary Risk Evaluation model in Europe [19]. In 2016, Gu et al. developed a sex-specific 10-year ASCVD risk prediction model based on Chinese people to predict the possibility of subjects having their first ASCVD event occurrence within 10 years using four large-scale prospective cohorts. The project used two cohorts to build the model and two others to validate the model’s efficacy, covering more than 100,000 people. These were the first ASCVD risk prediction models created specifically for the Chinese population, and the results proved to have excellent internal and external consistency [17]. Applying this tool would help identify high-risk populations and match the intensity of prevention interventions according to the absolute risk of individual ASCVD development.

Currently, several studies have focused on the association between CKD and CVD mortality. The Lancet once published that CKD and ESRD, particularly in individuals who also had proteinuria, were both explicit risk factors for increased CVD morbidity and mortality [20]. There are other studies indicating that decreased GFR is significantly associated with increased all-cause and CVD mortality [6,21], incident CHD in general people [22], and individuals with high-risk vascular events [23]. However, less attention has been paid to the association between CKD and ASCVD risk. Moreover, most studies concentrated on individuals with significantly reduced eGFR and end-stage renal failure (GFR < 60 mL/min/1.73 m^2^), while few focused on the risk of ASCVD in subjects with normal to slightly reduced kidney function.

Therefore, our study used a representative Chinese population sample to calculate eGFR based on the CKD-Epi equations, intending to explore the association between normal to mildly decreased eGFR and 10-year ASCVD.

## 2. Materials and Methods

### 2.1. Study Design and Setting

Data were obtained from the Chinese Physiological Constant and Health Condition (CPCHC) survey. This is a nationwide cross-sectional study covering multiple ethnic minorities, designed to find out the health condition and explore the general physiological constants of Chinese adults using a representative sample of the Chinese population. The study was conducted between 2007 and 2011 and collected a total of 13,239 eligible subjects from six provinces and autonomous regions of China. The detailed sampling procedures and quality control measurements of the study have been published previously [24,25].

### 2.2. Participants

All individuals who were willing to sign informed consent and voluntarily participate in the physical examination were included as study subjects. Subjects with heart, lung, liver, brain, kidney, or other major organ diseases, poor physical development, and symptoms of high fever or flu within the last 15 days were excluded from the project. In this study, subjects between the ages of 35 and 75 years were included, and individuals with missing primary study indicators (eGFR and 10-year ASCVD risk scores) were further excluded. A total of 12,986 subjects were ultimately included.

This study was approved by the Ethics Committee of the Institute of Basic Medical Sciences, Chinese Academy of Medical Sciences (Approved Number: 005-2008). Informed consent forms were obtained from each subject.

### 2.3. Data Collection

Demographic information, lifestyle factors, and health status were obtained from face-to-face interviews using standardized questionnaires. The physical examination and laboratory tests were performed by highly trained doctors and nurses following standard operating procedures. Data including subjects’ height, weight, waist circumference, and blood pressure will be collected through physical examination. Omron HEM-7000 electronic sphygmomanometers (Omron Healthcare; Muko, Kyoto, Japan) were used to measure blood pressure three times and the average values were taken as the final results. Venous blood was drawn from the subject to measure serum creatinine level, total cholesterol, and high-density lipoprotein cholesterol (HDL-C) to calculate the eGFRs and 10-year ASCVD risk scores. Blood samples were tested by Beckman AU series automatic biochemical analyzers (South Kraemer Blvd., Brea, CA, USA) and Sekisui Medical (Tokyo, Japan) reagents.

All names and identifiers were removed before any data were analyzed.

### 2.4. Estimation of GFR

Our study estimated GFRs using Chronic Kidney Disease Epidemiology Collaboration (CKD-EPI) 2009 equations, which are preferred in the United States and much of the world [14,16]. Many studies have shown that CKD-EPI equations are more accurate than the earlier MDRD ones, particularly for eGFR values greater than 60 mL/min/1.73 m^2^. The equations use age and serum creatinine levels to estimate GFR based on the subject’s race and sex. For Chinese adults, the CKD-EPI equation was expressed as a single equation:

GFR = 141 × min(Scr/κ, 1)α × max(Scr/κ, 1) − 1.209 × 0.993 Age × 1.018 [if female] 

Scr represented serum creatinine concentration, the unit of which was mg/dL. κ was 0.7 for females and 0.9 for males. α was −0.329 for females and −0.411 for males. Min indicates the minimum of Scr/κ or 1, and max indicates the maximum of Scr/κ or 1. Estimated GFR was reported in ml/minute/1.73 m^2^ of body surface area.

### 2.5. Assessment of 10-Year ASCVD Risk Level

The China-PAR project risk scores, which were established by Gu et al. [17], were used to assess the risk of subjects having their first ASCVD event occur within 10 years. The project built the sex-specific risk assessment models based on four large-scale contemporary population-based Chinese prospective cohorts and proved to have well-performed internal consistency and external validation. Details of the indicators and their parameters used in the model can be found [17]. Subjects were then divided into three risk level groups, low-risk group (10-year ASCVD risk score <5%), middle-risk group (risk score 5–10%), and high-risk group (risk score >10%), which were clinically and public health meaningful cut-off points.

### 2.6. Statistical Analysis

Continuous data were described as mean ± standard deviations (SDs), medians with interquartile ranges (IQRs), or 95% confidence intervals (95% CIs), as appropriate, and categorized data were summarized by numbers and percentages. With reference to the distribution of eGFR, eGFR was divided into four groups based on quartiles and medians to investigate the 10-year ASCVD risk outcomes in individuals with various eGFRs. Demographic and lifestyle factors among patients in different eGFR groups and ASCVD risk level groups were compared by one-way analysis of variance (ANOVA) or a chi-squared test. The eGFR was also compared among different ASCVD risk groups. We further conducted an ordinal logistic regression analysis to explore the association between 10-year ASCVD risk levels and eGFR. The model’s proportional odds assumption was satisfied according to a statistical test. Odds ratios (ORs) and their 95% CIs were calculated to express the strength of association. Two models were conducted to explore the association. Model 1 didn’t adjust any confounders, and Model 2 adjusted gender, race, occupation type, marital status, education level, drinking habits, and physical activity. Specifically, to avoid possible over-adjustment, those confounding variables that were used to construct the 10-year ASCVD risk score (including age, geographical area, residence in urban or rural village, smoking status, diabetes status, treated or untreated systolic blood pressure, waist circumference, total cholesterol, and family history of ASCVD) were not adjusted in the analysis.

All the analyses were two-sided, and a *p* value < 0.05 was considered significant. SAS 9.4 (SAS Institute Int., Cary, NC, USA) and R 4.1.3 were used for the statistical analysis.

## 3. Results

### 3.1. Baseline Characteristics of Participants

A total of 12,986 subjects were included in this study, excluding those individuals with missing eGFRs and ASCVD scores, which were both the main study indexes. Among them, 5976 (46.01%) were males and 7012 (53.99%) were females. Additionally, the majority of subjects were between 35 and 45 years old (33.04%), and individuals over 65 years of age represented the smallest proportion (13.99%). Every subject participated in questionnaire interviews, physical examinations, and laboratory tests to collect data. Data on demographic characteristics, laboratory results, and lifestyles are shown in Table 1.

### 3.2. eGFR

The baseline means eGFR was 92.54 ± 15.87 mL/min/1.73 m^2^. eGFRs were divided into four groups, Q1 (eGFR ≥ 104.4 mL/min/1.73 m^2^), Q2 (eGFR 93.6–104.4 mL/min/1.73 m^2^), Q3 (eGFR 81.6–93.6 mL/min/1.73 m^2^), Q4 (eGFR ≤ 81.6 mL/min/1.73 m^2^), respectively, based on eGFR quartiles. The above Table 1 summarized the demographic and lifestyle characteristics and laboratory test results of subjects among the quintiles of eGFR. The 10-year ASCVD risk scores showed an increasing trend as eGFR decreased, and the difference was statistically significant, suggesting that a decrease in eGFR might elevate the probability of ASCVD risk events within 10 years in subjects. On the other hand, eGFR also showed a correlation with age, as eGFR reduced with age (Table 1).

### 3.3. 10-Year ASCVD Risk 

According to Table 2, 4.4% of the subjects had 10-year ASCVD risk scores greater than 10%. The 10-year ASCVD risk increased progressively with increasing age, and the rate of 10-year ASCVD risk greater than 5% was much higher in male subjects than in females. Subjects who had smaller eGFR were more likely to have a greater risk of 10-year ASCVD. Additionally, the 10-year ASCVD risk was greater in individuals who consumed alcohol.

### 3.4. Association between eGFR and 10-Year ASCVD Risk Levels

Subjects in the Q2, Q3, and Q4 groups all had significantly higher 10-year ASCVD scores than those in the Q1 group (*p* < 0.001; Table 1). Meanwhile, compared with subjects’ eGFR in the low-risk group (mean eGFR: 94.36 mL/min/1.73 m^2^), the mean eGFR in the middle-risk (ASCVD risk score 5–10%) and high-risk (ASCVD risk score >10%) groups were 84.90 mL/min/1.73 m^2^ and 81.15 mL/min/1.73 m^2^, respectively, which were statistically significantly lower. The risk of one or higher-grade increase in the possibility of the first ASCVD event within 10 years in Q2, Q3, and Q4 was 5.95 times (95%CI: 4.72, 7.51), 12.35 times (95%CI: 9.86, 15.46) and 12.76 times (95%CI: 10.19, 15.97) higher compared with Q1, as shown in Table 3. Subsequently, we adjusted for the covariates including gender, race, occupation type, marital status, education level, drinking habits, and physical activity. Results indicated that the risk of one or more levels increasing in the 10-year ASCVD risk group was 5.20 times (Q2, 95%CI: 3.90, 6.94), 9.47 times (Q3, 95%CI: 7.15, 12.53) and 11.41 times (Q4, 95%CI: 8.61, 15.12) higher compared with Q1. In addition, all covariates showed strong associations with 10-year ASCVD risk, except for education level and alcohol consumption. Male, Han race, unregular physical activity, physical work, and being single or divorced were all risk factors for increased 10-year ASCVD risk.

## 4. Discussion

Our study found that the 10-year ASCVD risk scores gradually increased following the decrease in eGFR; the lower the subjects’ eGFR, the higher the 10-year ASCVD risk level they would have. Additionally, the association between eGFR and 10-year ASCVD risk level changed with varying eGFR. Associations with ASCVD risk levels were significantly higher in the Q2, Q3, and Q4 groups compared with the controlled group (eGFR ≥ 104.4 mL/min/1.73 m^2^), and the strength of the association gradually increased as eGFR decreased. Thus, in our population, the association of eGFR with 10-year ASCVD risk was linear.

Previous studies of patients with CVD or people at high risk for CVD have all generally been consistent in indicating an independent association between decreased kidney function and increased CVD outcomes [26,27,28]. However, this association has not been conclusively established for low or intermediate risk populations. The United States National Health and Nutrition Examination Survey (NHANES) and the Framingham study have obtained conflicting results. The level of kidney function was not found to be a risk factor for CVD outcomes in the Framingham cohort [29] or CVD death in the NHANES I [30]; however, was found to be a risk factor for CVD death in an analysis of the NHANES II [31]. Results from another community-based longitudinal study of coronary heart disease (CHD) and stroke in people aged 45–64 years showed that lower levels of kidney function were associated with a significantly increased risk of ASCVD over 5 years, and even normal to slightly reduced kidney function (60–89 mL/min/1.73 m^2^) was an independent risk factor for ASCVD and de novo ASCVD outcomes. After adjusting for traditional ASCVD risk factors as well as covariates, the estimated hazard ratio for ASCVD was 1.05 (95%CI: 1.02, 1.09) for every 10 mL/min/1.73 m^2^ reduction in eGFR [32]. Our study was consistent with the above findings in that we also found an independent association between kidney function levels and 10-year ASCVD risk. Associations with ASCVD risk levels were significantly higher in the Q2, Q3, and Q4 groups compared with the controlled group (eGFR ≥ 104.4 mL/min/1.73 m^2^), and the strength of the association gradually increased as eGFR decreased. Our results suggested that the association between eGFR and ASCVD risk score was linear at normal to moderate declines in kidney function (eGFR > 60 mL/min/1.73 m^2^), and that within a certain range (eGFR 60–93.6 in this study), the association was constant, which were also consistent with the results of a prior meta-analysis. The meta-analysis, which included more than 100,000 subjects, found that overall mortality and CVD mortality were fairly stable in the range of 60–120 mL/min/1.73 m^2^ and increased only at eGFR < 60 mL/min/1.73 m^2^ [33].

The reasons for the increased ASCVD risk due to reduced eGFR are still not clearly elucidated. There are several theoretical but unproven mechanisms. It has been found that most people with impaired renal function have a high prevalence of traditional or non-traditional ASCVD risk factors [5], such as older age, hypertension, diabetes, dyslipidemia [5], elevated homocysteine levels, and remnant cholesterol particles [32,34], which were all established risk factors for ASCVD [35,36,37]. Thus, patients with impaired renal function might be at heightened risk of developing ASCVD through traditional or non-traditional ASCVD risk factors. Second, kidney function insufficiency accelerated atherosclerosis. In the Atherosclerosis Risk in Communities (ARIC) Study, researchers demonstrated that for middle-aged, non-cardiovascular high-risk subjects, a lower eGFR level was associated with a marked increase in the prevalence of ASCVD over five years, proving that GFR levels were independent risk factors for ASCVD [32]. Inflammation and oxidative stress were possible reasons to account for the association [34]. Moreover, reduced eGFR levels were markers of renal function insufficiency, which could lead to decreased clearance of metabolic waste products in the body, causing accumulation of ASCVD serum markers in the body and a variety of metabolic abnormalities [38,39]. Finally, the interactions between the heart and kidneys were multiple and complex, with primary dysfunction in one of the heart and kidneys often leading to secondary impairment in the other, a phenomenon known as cardiorenal syndrome (CRS), and the reduced renal function itself might be a marker for the progression of cardiac insufficiency [40].

Additionally, we divided the eGFR into four groups based on the quartiles and medians of eGFR, since our study population was composed of subjects with mild to moderate declines in renal function but still in relatively healthy status, which would result in an uneven distribution of people in the different groups if eGFR ≥ 90 mL/min/1.73 m^2^ (Kidney damage with normal or increased GFR) and eGFR 60–89 mL/min/1.73 m^2^ (Kidney damage with mildly decreased GFR) were used as cut-off points according to the clinical guidelines published by the National Kidney Foundation [10]. Moreover, considering the practical application value of the study, we chose quartiles for grouping instead of triple quartiles.

Our study differed from previous studies in many aspects. Firstly, we used CKD-Epi equations to calculate eGFR due to their accuracy in subjects with normal to mildly reduced renal function, when compared to MDRD formulas. Secondly, we conducted the study on a relatively homogeneous healthy population, excluding subjects with histories of issues with the heart, brain, liver, kidneys and other important organs, poor physical development, and those with high fever or influenza symptoms within the past 15 days, thus avoiding the confounding effect of factors that have a large impact on cardiovascular outcomes, such as diabetes and renal failure, on the study index. Thirdly, the 10-year ASCVD risk score model was established based on four prospective, large-scale cohorts covering over 100,000 Chinese people with over 10 years of follow-up, showing excellent internal and external consistency. Our study was the first to use this tool to explore the association between eGFR and ASCVD risk. Additionally, we used a representative sample of the whole Chinese population, covering subjects of various ages, genders, and races so that the results could reflect the actual situation of the Chinese population.

Our study also had certain limitations. First of all, it was a cross-sectional study, lacking the ability to make the causal inference. Moreover, as our study included subjects in relatively healthy conditions, fewer had eGFR < 60 mL/min/1.73 m^2^, thus limiting the ability to explore the association between individuals with significantly decreased eGFR (eGFR < 60 mL/min/1.73 m^2^) and 10-year ASCVD risk. However, there have been several studies confirming that a significantly decreased eGFR is an established risk factor for CVD [6,22,41]; therefore, we could infer their associations with ASCVD to some extent.

## 5. Conclusions

In conclusion, among Chinese subjects with normal to slightly reduced kidney function, we found that eGFR is significantly associated with 10-year ASCVD risk, even after adjusting for the covariates. Therefore, clinicians should strengthen the monitoring of cardiovascular risk in patients with renal impairment (even if renal function is only mildly reduced), assess the patients’ risk of ASCVD and take early action in high-risk groups to reduce the risk of adverse cardiovascular events.

## Figures and Tables

**Table 1 ijerph-19-16300-t001:** Baseline characteristics of participants.

Variables	Q1, ≥104.4 mL/min/1.73 m^2^	Q2, 93.6–104.4 mL/min/1.73 m^2^	Q3, 81.6–93.6 mL/min/1.73 m^2^	Q4, ≤81.6 mL/min/1.73 m^2^	*p*
	*n* = 3252	*n* = 3230	*n* = 3255	*n* = 3249	
eGFR (mL/min/1.73 m^2^)	111.80 ± 5.76	98.97 ± 3.10	87.96 ± 3.48	71.43 ± 8.51	<0.001
Scr (mg/dL)	0.66 ± 0.12	0.76 ± 0.12	0.87 ± 0.13	1.02 ± 0.19	<0.001
10-Year ASCVD risk scores (%)	0.65 (0.30, 1.42)	2.69 (0.90, 3.52)	2.23 (0.79, 5.10)	2.37 (0.90, 5.24)	<0.001
10-Year ASCVD risk level					<0.001
Low risk	3163 (97.26)	2763 (85.54)	2418 (74.29)	2401 (73.90)	
Middle risk	75 (2.31)	391(12.11)	617 (18.96)	592 (18.22)	
High risk	14 (0.43)	76 (2.35)	220 (6.76)	256 (7.88)	
Sex					<0.001
Male	1339 (22.41)	1531 (25.62)	1668 (27.91)	1438 (24.06)	
Female	1915 (27.31)	1699 (24.23)	1587 (22.63)	1811 (25.83)	
Age group					<0.001
35–44 years	2074 (48.33)	700 (16.31)	909 (21.18)	608 (14.17)	
45–54 years	1029 (27.35)	1176 (31.26)	803 (21.35)	754 (20.04)	
55–64 years	137 (4.40)	1170 (37.55)	777 (24.94)	1032 (33.12)	
65–75 years	12 (0.66)	184 (10.13)	766 (42.16)	855 (47.06)	
Education					<0.001
Primary School and blow	429 (11.83)	715 (19.72)	1051 (28.99)	1431 (39.46)	
Middle School	1502 (26.96)	1503 (26.97)	1361 (24.43)	1206 (21.64)	
University and above	1230 (36.32)	894 (26.40)	743 (21.94)	520 (15.36)	
Unknown	93	118	100	92	
Marriage					<0.001
Married	3070 (26.50)	2841 (24.53)	2856 (24.67)	2814 (24.30)	
Single	37 (28.03)	34 (25.76)	33 (25.00)	28 (21.21)	
Divorced or widowed	102 (14.23)	180 (25.10)	212 (29.57)	223 (31.10)	
Unknown	45	175	154	184	
Drink					<0.001
No	2373 (24.61)	2333 (24.20)	2408 (24.97)	2528 (26.22)	
Yes	881 (26.33)	897 (26.81)	847 (25.31)	721 (21.55)	
Smoke					0.002
No	2434 (25.91)	2311 (24.60)	2284 (24.32)	2364 (25.17)	
Yes	820 (22.81)	919 (25.56)	971 (27.01)	885 (24.62)	
Occupation					<0.001
Physical job	1452 (17.62)	1937 (23.51)	2288 (27.77)	2562 (31.10)	
Mental job	1802 (37.94)	1293 (27.23)	967 (20.36)	687 (14.47)	
Exercise					<0.001
No	1816 (26.19)	1556 (22.44)	1728 (24.92)	1834 (26.46)	
Yes	684 (18.95)	956 (26.48)	950 (26.32)	1020 (28.25)	
Unknown	754	718	577	395	
Urbanization					<0.001
Suburban and rural areas	2314 (28.83)	2090 (26.04)	1921 (23.93)	1701 (21.20)	
Urban areas	940 (18.94)	1140 (22.97)	1334 (26.88)	1548 (31.20)	
Race					<0.001
Han	2158 (26.05)	2237 (27.00)	2016 (24.34)	1873 (22.61)	
Other	1096 (23.30)	993 (21.11)	1239 (26.34)	1376 (29.26)	
Province					<0.001
Sichuan	630 (32.44)	585 (30.12)	443 (22.81)	284 (14.62)	
Heilongjiang	627 (30.38)	637 (30.86)	499 (24.18)	301 (14.58)	
Hunan	795 (43.16)	568 (30.84)	317 (17.21)	162 (8.79)	
Inner Mongolia	1002 (38.70)	830 (32.06)	519 (20.05)	238 (9.20)	
Yunnan	70 (2.68)	256 (9.81)	827 (31.69)	1457 (55.82)	
Ningxia	130 (6.70)	354 (18.24)	650 (33.49)	807 (41.58)	

eGFR, estimated glomerular filtration rate; Scr, serum creatinine concertation; ASCVD, Atherosclerotic Cardiovascular Disease.

**Table 2 ijerph-19-16300-t002:** Demographic and lifestyle factors in subjects by 10-Year ASCVD risk group.

Variables	Low-Risk	Middle-Risk	High-Risk	*p*
	*n* = 10,747	*n* = 1675	*n* = 566	
eGFR (mL/min/1.73 m^2^)	94.36 ± 15.80	84.90 ± 13.80	81.15 ± 14.22	<0.001
Sex				<0.001
Male	4134 (69.18)	1319 (22.07)	523 (8.75)	
Female	6613 (94.31)	356 (5.08)	43 (0.61)	
Age group				<0.001
35–44 years	4235 (98.65)	53 (1.23)	5 (0.12)	
45–54 years	3482 (92.56)	238 (6.33)	42 (1.12)	
55–64 years	2332 (74.84)	632 (20.28)	152 (4.88)	
65–75 years	698 (38.41)	752 (41.39)	367 (20.20)	
eGFR (mL/min/1.73 m^2^)				<0.001
≤81.6	2401(73.90)	592 (18.22)	256 (7.88)	
81.6–93.6	2418 (74.29)	617 (18.96)	220 (6.76)	
93.6–104.4	2763 (85.54)	391 (12.11)	76 (2.35)	
≥104.4	3163 (97.26)	75 (2.30)	14 (0.43)	
Education				<0.001
Primary School and blow	2997 (82.65)	450 (12.41)	179 (4.94)	
Middle School	4502 (80.80)	806 (14.47)	264(4.74)	
University and above	2944 (86.92)	343 (10.13)	100(2.95)	
Unknown	304	76	23	
Marriage				<0.001
Married	9745 (84.15)	1374(11.86)	462(3.99)	
Single	107 (81.06)	15(11.36)	10(7.58)	
Divorced or widowed	548 (76.43)	138(19.25)	31(4.32)	
Unknown	347	148	63	
Drink				<0.001
No	8270 (85.77)	1006 (10.43)	366 (3.80)	
Yes	2477 (74.03)	669 (19.99)	200 (5.98)	
Occupation				<0.001
Physical job	6581 (79.88)	1221 (14.82)	437 (5.30)	
Mental job	4166 (87.72)	454 (9.56)	129 (2.72)	
Exercise				<0.001
No	6081 (87.70)	644 (9.29)	209 (3.01)	
Yes	2727 (75.54)	632 (17.51)	251 (6.95)	
Unknown	1939	399	106	
Race				<0.001
Han	6777 (81.81)	1158 (13.98)	349 (4.21)	
Other	3970 (84.40)	517 (10.99)	217 (4.61)	

eGFR, estimated glomerular filtration rate; ASCVD, Atherosclerotic Cardiovascular Disease.

**Table 3 ijerph-19-16300-t003:** Association of eGFR and 10-Year ASCVD risk levels using ordinal logistic model.

	Q1, ≥104.4 mL/min/1.73 m^2^ *n* = 3252	Q2, 93.6–104.4 mL/min/1.73 m^2^ *n* = 3230	Q3, 81.6–93.6 mL/min/1.73 m^2^ *n* = 3255	Q4, ≤81.6 mL/min/1.73 m^2^ *n* = 2934
Model 1	1 (reference)	5.95 (4.72, 7.51)	12.35 (9.86, 15.46)	12.76 (10.19, 15.97)
Model 2	1 (reference)	5.20 (3.90, 6.94)	9.47 (7.15, 12.53)	11.41 (8.61, 15.12)

Model 1 didn’t adjust any confounders; Model 2 further adjusted gender, race, occupation type, marital status, education level, drinking habits, and physical activity.

## Data Availability

The data underlying this article will be shared on reasonable request to the corresponding author (email: xutaosd@ibms.pumc.edu.cn).

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
