# Peer review of "Association between 10-Year Atherosclerotic Cardiovascular Disease Risk and Estimated Glomerular Filtration Rate in Chinese People with Normal to Slightly Reduced Kidney Function: A Cross-Sectional Study"

_ijerph, 2022, doi:10.3390/ijerph192316300_

Round 1

Reviewer 1 Report

This paper presents an association analysis regarding the estimated glomerular filtration rate and 10-Year Atherosclerotic Cardiovascular Disease Risk. Overall, the analysis is noisy and the results may be overwhelmed by confounding factors with a high risk. Detailed comments are as follows.

1. Abbreviations in the abstract should be explained, such as ASCVD, Q1. 

2. Abstract should stand alone. The background, question formulation and basic explanation of the most significant results are absent.

3. Introduction: The motivation for analyzing the association between eGFR and ASCVD is insufficient. There should be a physiological or phenomenological indicator presented by previous studies to motivate such an analysis.

4.  The whole analysis does not consider any confounding factors and the group strategy seems unreasonable. How to distinguish the analyzed association is not affected by a common cause or a controlled common consequence? Why divided eGFR in such four groups? Moreover, it would be beneficial to first present a factor graph that shows the potential associations among GFR, ASCVD and other factors and then control each factor to analyze the significance. 

5. considering the blurry analysis, the discussion and conclusion is not applicable to be judged and rephrased after the amended analysis. 

Author Response

Dear reviewer,

Many thanks for your comments and advices. These are my responses to the comments.

This paper presents an association analysis regarding the estimated glomerular filtration rate and 10-Year Atherosclerotic Cardiovascular Disease Risk. Overall, the analysis is noisy and the results may be overwhelmed by confounding factors with a high risk. Detailed comments are as follows.

  1. Abbreviations in the abstract should be explained, such as ASCVD, Q1. 

Response:

Many thanks for your comment. We have explained the meaning of the abbreviations in the summary as you requested. ASCVD stands for “Atherosclerotic Cardiovascular Disease”, and Q1, Q2, Q3, Q4 represent four sets of data divided by quartiles and medians, which were Quartile 1, Quartile 2, Quartile 3 and Quartile 4.

  1. Abstract should stand alone. The background, question formulation and basic explanation of the most significant results are absent.

Response: Many thanks for your comment. We have added contents to the abstract section to make it more Integral. Detailed Contents as follows:

“Many studies suggest that cardiovascular-related mortality was higher in patients with end-stage renal disease, but few focused on the risk of atherosclerotic cardiovascular disease (ASCVD) in subjects with normal to slightly reduced kidney function. Our study aimed to explore the association between normal to slightly decreased estimated glomerular filtration rate (eGFR) and 10-year ASCVD risk levels among subjects in relative health conditions. 12,986 subjects from Chinese Physiological Constant and Health Condition survey were included. We used the Chronic Kidney Disease Epidemiology Collaboration equations to calculate eGFR and the 10-year ASCVD risk scores to assess subjects’ risk of 10-Year ASCVD. Ordinal logistic regression analysis was conducted to explore the association between ASCVD risk levels and eGFR and adjust the possible confounding factors. Results indicated that the 10-year ASCVD risk scores gradually increased following the decrease of eGFR. Additionally, the association between eGFR and 10-year ASCVD risk level changed with varying eGFR. The risk of one or more levels increasing in the 10-year ASCVD risk group was 5.20 times (Quartile 2 (Q2), 95%CI: 3.90, 6.94), 9.47 times (Q3, 95%CI: 7.15, 12.53) and 11.41 times (Q4, 95%CI: 8.61, 15.12) higher compared with Q1. We found that eGFR was significantly associated with 10-year ASCVD risk among Chinese subjects with normal to slightly reduced kidney function. Therefore, clinical doctors should enhance monitoring of the renal function of high-risk ASCVD subjects and ASCVD patients and take early steps to intervene in renal insufficiency to reduce the risk of adverse events.”

  1. Introduction: The motivation for analyzing the association between eGFR and ASCVD is insufficient. There should be a physiological or phenomenological indicator presented by previous studies to motivate such an analysis.

Response: Many thanks for your comment.

Indeed, we did not fully explain the motivation for this study in the introduction section. Thus we have included relevant researches findings in the first paragraph of the introduction to fully explain the association between decreased kidney function and cardiovascular disease. As estimated glomerular filtration rate was an important indicator of renal function, it is significant to examine the association between estimated glomerular filtration rate and cardiovascular disease. In previous studies, numerous investigators have focused on the strong association between end-stage kidney disease and cardiovascular events, and the importance of strengthening cardiovascular event prevention interventions in patients with end-stage kidney disease has already been highlighted in guidelines. However, fewer studies have focused on the risk of cardiovascular disease in subjects with mildly reduced renal function.

All of the above are described in detail in our introduction section. In addition, the physiological mechanisms of reduced renal function leading to cardiovascular disease are briefly described in the Discussion section and are therefore not presented in detail in the Introduction section.

The following has been added to the introduction section:

“Several studies both indicated that patients with ESRD have a CVD mortality that was 10 to 20 times greater than the general population(1-3). What’s more, the National Kidney Foundation (NKF) Task Force on Cardiovascular Disease in Chronic Renal Disease issued a report in 1998, emphasizing that the high risk of CVD in patients with CKD (4). This report showed that there was a high prevalence of CVD in CKD and that mortality due to CVD was 10 to 30 times higher in dialysis patients than in the general population (5, 6). The task force therefore recommended that patients with CKD should be considered in the “highest risk group” for subsequent CVD events and that treatment based on CVD risk stratification should take into account the highest-risk status of patients with CKD (2).”

  1. The whole analysis does not consider any confounding factors and the group strategy seems unreasonable. How to distinguish the analyzed association is not affected by a common cause or a controlled common consequence? Why divided eGFR in such four groups? Moreover, it would be beneficial to first present a factor graph that shows the potential associations among GFR, ASCVD and other factors and then control each factor to analyze the significance. 

Response: Many thanks for your comment.

  • About the confounding factors. Sorry for the misunderstanding. We did adjust some of the covariates in our analysis, but we have not made this clear in the 'Statistical analysis' section, so we have added the explanation about the confounding factors as you suggested.

In this study, associations between slightly to mild reduced eGFR and 10-year ASCVD risk levels were explored using ordinal logistic regression models. Two models were conducted to explore the association. Model 1 didn’t adjust any confounders, and Model 2 adjusted gender, race, occupation type, marital status, education level, drinking habits, and physical activity. These covariates were selected based on variables with clear confounding effects found in previous studies and common demographic characteristic factors. Specifically, to avoid possible overadjustment, those variables that were used to construct the 10-year ASCVD risk score (including age, geographical area, residence in urban or rural village, smoking status, diabetes status, treated or untreated systolic blood pressure, waist circumference, total cholesterol, and family history of ASCVD) were not adjusted in the analysis.

  • About the eGFR classification. We divided the eGFR into four groups based on the quartiles and medians of eGFR, because our study population was composed of subjects with mild to moderate declines in renal function but in a relatively healthy status, which would result in an uneven distribution of numbers in the different groups if eGFR >= 90 (Kidney damage with normal or increased GFR) and eGFR 60-89 (Kidney damage with mildly decreased GFR) were used as cut-off points according to the clinical guidelines published by the National Kidney Foundation(7) . Secondly, considering the practical application value of the study, we chose quartiles for grouping instead of triple quartiles. Finally, we used this reasonable grouping of independent variables eGFR with reference to previous studies (8-10).
  1. considering the blurry analysis, the discussion and conclusion is not applicable to be judged and rephrased after the amended analysis. 

 Response: Many thanks for your comment. We have restated the parts that caused your misunderstanding. Our study has adjusted common covariates and made explanation for specific variables that were not adjusted. In addition, we also used a reasonable grouping of independent variables eGFR with reference to previous studies. We therefore hope that you will re-evaluate our study.

Thank you for your kind attention and looking forwards to your favorable reply.

Yours sincerely,

Feilong Chen

Tao Xu

Department of epidemiology and statistics

Institute of Basic Medical Sciences, Chinese Academy of Medical Sciences & School of Basic Medicine, Peking Union Medical College

5, Dong dan san tiao

Beijing 100005

China

Tel: 86 10 69156408

  1. Levey AS, Coresh J, Balk E, Kausz AT, Levin A, Steffes MW, et al. National Kidney Foundation practice guidelines for chronic kidney disease: evaluation, classification, and stratification. Annals of internal medicine (2003) 139(2):137-47. Epub 2003/07/16. doi: 10.7326/0003-4819-139-2-200307150-00013. PubMed PMID: 12859163.
  2. Sarnak MJ, Levey AS, Schoolwerth AC, Coresh J, Culleton B, Hamm LL, et al. Kidney disease as a risk factor for development of cardiovascular disease: a statement from the American Heart Association Councils on Kidney in Cardiovascular Disease, High Blood Pressure Research, Clinical Cardiology, and Epidemiology and Prevention. Hypertension (Dallas, Tex : 1979) (2003) 42(5):1050-65. Epub 2003/11/08. doi: 10.1161/01.HYP.0000102971.85504.7c. PubMed PMID: 14604997.
  3. Weiner DE, Tighiouart H, Amin MG, Stark PC, MacLeod B, Griffith JL, et al. Chronic kidney disease as a risk factor for cardiovascular disease and all-cause mortality: a pooled analysis of community-based studies. Journal of the American Society of Nephrology : JASN (2004) 15(5):1307-15. Epub 2004/04/22. doi: 10.1097/01.asn.0000123691.46138.e2. PubMed PMID: 15100371.
  4. Levey AS, Beto JA, Coronado BE, Eknoyan G, Foley RN, Kasiske BL, et al. Controlling the epidemic of cardiovascular disease in chronic renal disease: what do we know? What do we need to learn? Where do we go from here? National Kidney Foundation Task Force on Cardiovascular Disease. American journal of kidney diseases : the official journal of the National Kidney Foundation (1998) 32(5):853-906. Epub 1998/11/20. doi: 10.1016/s0272-6386(98)70145-3. PubMed PMID: 9820460.
  5. Foley RN, Parfrey PS, Sarnak MJ. Clinical epidemiology of cardiovascular disease in chronic renal disease. American journal of kidney diseases : the official journal of the National Kidney Foundation (1998) 32(5 Suppl 3):S112-9. Epub 1998/11/20. doi: 10.1053/ajkd.1998.v32.pm9820470. PubMed PMID: 9820470.
  6. Manyari DE. Prognostic implications of echocardiographically determined left ventricular mass in the Framingham Heart Study. The New England journal of medicine (1990) 323(24):1706-7. Epub 1990/12/13. doi: 10.1056/nejm199012133232413. PubMed PMID: 2146505.
  7. (KDIGO) KDIGO. KDIGO clinical practice guideline for the evaluation and management of chronic kidney disease. Kidney International Supplements (2013) 3:1-150.
  8. Yoshikawa D, Ishii H, Suzuki S, Takeshita K, Kumagai S, Hayashi M, et al. Plasma indoxyl sulfate and estimated glomerular filtration rate. Circulation journal : official journal of the Japanese Circulation Society (2014) 78(10):2477-82. Epub 2014/08/12. doi: 10.1253/circj.cj-14-0401. PubMed PMID: 25109428.
  9. Fauchier L, Bisson A, Clementy N, Vourc'h P, Angoulvant D, Babuty D, et al. Changes in glomerular filtration rate and outcomes in patients with atrial fibrillation. American heart journal (2018) 198:39-45. Epub 2018/04/15. doi: 10.1016/j.ahj.2017.12.017. PubMed PMID: 29653646.
  10. Karhapää P, Pihlajamäki J, Pörsti I, Kastarinen M, Mustonen J, Niemelä O, et al. Glomerular filtration rate and parathyroid hormone are associated with 1,25-dihydroxyvitamin D in men without chronic kidney disease. Journal of internal medicine (2012) 271(6):573-80. Epub 2011/10/15. doi: 10.1111/j.1365-2796.2011.02471.x. PubMed PMID: 21995281.

Reviewer 2 Report

The study aims to explore the association between mildly reduced eGFR to normal and 10-year atherosclerotic vascular disease (nonfatal acute myocardial infarction or coronary heart disease death or fatal or nonfatal stroke).

Subjects from the Chinese Physiological Constant and Health Condition survey without major organ diseases were included. The design was a cross-sectional study. The statistical analysis should be better described, in particular, the association between 10-year ASCVD risk level (three groups) with eGFR was modelled with logistic regression, does that mean an ordinal logistic regression or a multinomial regression?

The results are clearly reported and the discussion is exhaustive.

Author Response

Dear reviewer,

Many thanks for your comments and advices. These are my responses to the comments.

“The study aims to explore the association between mildly reduced eGFR to normal and 10-year atherosclerotic vascular disease (nonfatal acute myocardial infarction or coronary heart disease death or fatal or nonfatal stroke)”.

  1. Subjects from the Chinese Physiological Constant and Health Condition survey without major organ diseases were included. The design was a cross-sectional study. The statistical analysis should be better described, in particular, the association between 10-year ASCVD risk level (three groups) with eGFR was modelled with logistic regression, does that mean an ordinal logistic regression or a multinomial regression?

Response:

 Thank you for your very valuable review. 

I am sorry for your misunderstanding due to my unclear statement. We use the ordinal logical regression model in the analysis because the dependent variable, 10-year ASCVD risk level, is an ordinal three classified variables: low risk (10-year ASCVD risk score<5%), medium risk (risk score 5%-10%) and high risk (risk score >10%), and the data was statistically tested to be satisfied with the model's proportional odds assumption. I have revised this part of the presentation in the methodology section to remove misunderstandings.

  1. The results are clearly reported and the discussion is exhaustive.

Response:

Thank you very much for your affirmation.

Thank you for your kind attention and looking forwards to your favorable reply.

Yours sincerely,

Feilong Chen

Tao Xu

Department of epidemiology and statistics

Institute of Basic Medical Sciences, Chinese Academy of Medical Sciences & School of Basic Medicine, Peking Union Medical College

5, Dong dan san tiao

Beijing 100005

China

Tel: 86 10 69156408

Round 2

Reviewer 1 Report

The authors address all my concerns. Thank for the authors efforts and congratulations on their work.